# Quality Improvement of Zhayu, a Fermented Fish Product in China: Effects of Inoculated Fermentation with Three Kinds of Lactic Acid Bacteria

**DOI:** 10.3390/foods11182756

**Published:** 2022-09-08

**Authors:** Yueqi An, Xiaowen Cai, Lin Cong, Yang Hu, Ru Liu, Shanbai Xiong, Xiaobo Hu

**Affiliations:** 1College of Food Science and Technology/National R&D Branch Center for Conventional Freshwater Fish Processing (Wuhan), Huazhong Agricultural University, Wuhan 430070, China; 2Engineering Research Center of Green Development for Conventional Aquatic Biological Industry in the Yangtze River Economic Belt, Ministry of Education, Wuhan 430070, China

**Keywords:** fermented fish product, lactic acid bacteria, inoculated fermentation, quality improvement, flavor characteristics

## Abstract

To investigate the effects of inoculation fermentation on the quality of Zhayu (a traditional fermented fish product in China), different amounts of *L. plantarum*, *P. acidilactici*, and *P. pentosaceus* were inoculated into samples, and the safety, nutritional, textural, and flavor properties of the samples were evaluated. Fermentation with lactic acid bacteria (LAB) decreased pH values and total volatile basic nitrogen content. The addition of 10^8^~10^9^ cfu/100 g LAB significantly increased the content of crude fat and water-soluble proteins in Zhayu. The addition of *L. plantarum* and *P. acidilactici* increased the content of soluble solids in Zhayu. Moreover, fermentation with LAB made the products tender and softer, and the samples prepared with 10^9^ cfu/100 g LAB presented better overall qualities. Additionally, Zhayu fermented with *L. plantarum* and *P. acidilactici* showed the strongest sourness, while the samples prepared with *P. pentosaceus* showed the strongest umami taste, consistent with the highest contents of Asp (25.1 mg/100 g) and Glu (67.8 mg/100 g). The addition of LAB decreased the relative contents of aliphatic aldehydes, (Z)-3-hexen-1-ol, and 1-octen-3-ol, reducing the earthy and fishy notes. However, LAB enhanced the contents of terpenoids, acids, esters, and S-containing compounds, increasing the sour, pleasant, and unique odors of Zhayu.

## 1. Introduction

Fermentation is one of the main ways of the deep processing and preservation of aquatic products [1]. Fermented aquatic products have been favored by consumers in various regions because of their unique texture and flavor, such as Thai Plaa-som [2], Indian Bakasang [3], Indian Ngari [4], Chinese Suanyu [5,6,7], and Chinese fermented mandarin fish [8,9]. Different from other fermented fish products (such as Suanyu), Zhayu is fermented in small pieces with rice flour [10]. Zhayu is usually prepared with freshwater fish, especially grass carp (*Ctenopharyngodon idellus*). To manufacture Zhayu, washed fish fillets are cut into cubes and mixed with rice flour, salt, hot pepper, ginger, and other seasonings, and then the mixture is sealed and fermented under a solid state (Figure 1). During the solid-state fermentation, proteolysis, lipid degradation, and carbohydrate decomposition actioned by enzymes and microorganisms result in the production of organic acids, amino acids, small molecular peptides, and other substances, which enhance the nutritional value, enrich the flavor characteristics, and leads to a unique soft and loose texture of the fermented fish products [5,11]. In recent years, fermented Zhayu products have shown good market potential due to their delicious flavor and high nutritional value.

Traditional fermented fish products were usually manufactured by spontaneous fermentation without the addition of starter cultures in small-scale processing units. However, the conditions of spontaneous fermentation were difficult to control, resulting in unequal qualities of fermented fish products, which was not conducive to their industrial production. To solve this problem, yeast, mold, and bacteria (especially lactic acid bacteria and staphylococci) have been used as starter cultures for fermented fish products [12,13,14]. Fermentation with starter cultures not only shortens the fermentation time and prolongs the storage time of fermented products, but also improves the flavor, color, safety, and nutritional qualities of the products [15,16]. Lactic acid bacteria (LAB), especially *Lactiplantibacillus*
*plantarum* and *Pediococcus pentosaceus*, have been isolated and identified as the dominant strains in Zhayu [17]. Moreover, *Pediococcus acidilactici* is also an important LAB in fermented fish or meat products [13,18]. LAB can reduce the pH value of the fermented product and inhibit the growth of spoilage microorganisms in the product [6,15]. LAB can also decrease the content of nitrite in fermented aquatic products and improve the safety quality of the product [19]. Moreover, LAB has been shown to make fermented sausage show better flavor properties [20]. Therefore, LAB could be used as a starter culture of Zhayu to improve its qualities. However, the effects of LAB species and inoculation amounts on the improvement of the qualities of Zhayu remain to be further studied. 

In the present study, fermented Zhayu samples were prepared by inoculation with different amounts of *Lactiplantibacillus*
*plantarum* (*L**. plantarum*), *Pediococcus acidilactici* (*P.*
*acidilactici*), and *Pediococcus pentosaceus* (*P.*
*pentosaceus*) for starter culture fermentation, and the Zhayu sample prepared with natural fermentation was taken as the control. All Zhayu samples were fermented in the same conditions. The safety, nutritional, textural, and flavor properties of these fermented fish products were evaluated. The objectives of this research were (1) to investigate the effects of inoculation amounts and LAB species on the Zhayu products’ quality and (2) to compare the effects of different kinds of LAB on the quality improvement of the fermented Zhayu products. This study will provide some theoretical basis for the production and processing of fermented aquatic products and give some new ideas for promoting industrialization and large-scale production of fermented aquatic products.

## 2. Materials and Methods

### 2.1. Materials and Chemicals

Live grass carp (*Ctenopharyngodon idellus*) (approximately 1500~2000 g) were purchased from the market of Huazhong Agricultural University (Wuhan, Hubei, China) and transported to the laboratory within 15 min in a transport case with water to keep them alive. 

Rice was purchased from the market of Huazhong Agricultural University (Wuhan, Hubei, China) and smashed (<60 mesh) before use. Salt, hot pepper, and ginger were of food grade and were purchased from Zhongbai supermarket (Wuhan, Hubei, China).

The internal standard (2-octanol, >99.5%) used for gas chromatography–mass spectrometry (GC-MS) analysis was supplied by Sigma-Aldrich (St. Louis, MO, USA). The standard, reagent, and eluent solvent used for the amino acid analysis were supplied from Waters Corporation (Milford, MA, USA). All other chemicals were of analytical grade and were purchased from Sinopharm Chemical Reagent Co., Ltd. (Shanghai, China).

### 2.2. Preparation of Starter Culture

*L. plantarum* (CCTCC NO. M 2012396) was obtained from the China Center for Type Culture Collection (Peking, China). *P. acidilactici* (CICC 10344) and *P. pentosaceus* (CICC 22227) were purchased from the China Center of Industrial Culture Collection (Peking, China). These LAB were separately subcultured twice in DeMan Rogosa Sharpe (MRS) agar (Hopebio Co. Ltd., Qingdao, Shandong, China) and cultured at 30 °C for 48 h. Then, LAB was incubated in 200 mL of MRS broth (Hopebio Co. Ltd., Qingdao, Shandong, China). After the amplification of culture, the mixture of strains and MRS broth was centrifuged at 8000 r/min for 15 min at 4 °C. Cell pellets were harvested and washed with physiological saline (0.9% NaCl, *w*/*v*) twice. Then, cell pellets were resuspended with 0.9% NaCl (*w*/*v*). The growth curves of the three strains were measured by a UV-1700 ultraviolet spectrophotometer (Shimadzu, Kyoto, Japan) at 600 nm [7], and the central time point of the logarithmic phase was considered as the amplification time required for the strain’s amount around 10^9^ cfu/mL. Then, the strains were harvested for the exact time and the plate count agar was used to determine and verify the amount of strains. The bacterial suspension (10^9^ cfu/mL) was diluted to 10^8^ cfu/mL, 10^7^ cfu/mL, and 10^6^ cfu/mL in 0.9% NaCl (*w*/*v*) for the followed inoculation. Each bacterial suspension was stored at 4 °C and used within 24 h. 

### 2.3. Sampling

Live grass carp were killed by a physical blow to the head, beheaded, gutted, and cleaned, following the guidance on Treating Experimental Animals developed by China’s Ministry of Science and Technology in 2006 and regulations issued by the China State Council in 1988. Grass carp were cut into small pieces (approximately 1 cm × 2 cm × 2 cm). Seeds of hot pepper and ginger peals were removed, and the hot pepper and ginger were chopped. Fish pieces (100 g) were mixed in a vacuum bag (food grade) with rice flour (45 g), chopped hot pepper (20 g), bruised ginger (8 g), salt (8 g), tap water (45 mL), and different species of LAB suspension (1 mL). After mixing, the bag was vacuumed and sealed by a vacuum packaging machine. The control sample was prepared by spontaneous fermentation without the addition of starter cultures. After sealing, the mixtures were fermented at 25 °C for 60 h. Before analysis, the rice flour and seasoning were removed from the Zhayu samples. 

### 2.4. Determination of pH, Titratable Acidity (TA), and Total Volatile Basic Nitrogen (TVB-N)

The pH values of samples were determined as reported by Zeng et al. [15] with some modifications. Ten grams of samples were homogenized with 50 mL of deionized water (discharged CO_2_) at 8000 r/min for 2 min. After vacuum filtration, the pH values of filtrates were measured by a digital pH meter (Mettler Toledo FE28, Shanghai, China).

TA was determined via AOAC [21], and the results were expressed as lactic acid content (mg/g). TVB-N contents were determined by the micro-diffusion method of Conway [22].

### 2.5. Determination of Crude Fat, Water-Soluble Protein, and Soluble Solids

Crude fat contents were determined by an automated Soxhlet method [23] with some modifications. Two grams of homogenized and dried samples were extracted with petroleum ether (boiling point: 30~60 °C) rather than diethyl ether. 

Water-soluble protein was extracted according to the method of Liao et al. [24] with some modifications. Ten grams of samples were homogenized with 0.05 mol/L phosphate buffer (pH 7.0) and diluted to 100 mL. After centrifuging at 8000 r/min for 20 min at 4 °C, the protein contents in the supernatant were measured by the Lowry method [25]. 

To determine the content of soluble solids, 10 g of samples was homogenized with distilled water and diluted to 100 mL. After centrifuging at 8000 r/min for 20 min at 4 °C, 10 g of the supernatant was dried at 105 ± 2 °C. The content of soluble solids was expressed as the percentage of dry matter weight.

### 2.6. Texture Analysis

Textural profile analysis (TPA) was carried out using a model TA-XT Plus texture analyzer (Stable Micro System, Surrey, UK) at room temperature (22 ± 2 °C). The Zhayu sample was placed on the test platform with the 2 cm × 2 cm side up. The probe type for the test was P/36R, the pre-measurement speed was 2.00 mm/s, the measurement speed was 1.00 mm/s, the post-measurement speed was 5.00 mm/s, and the compression strain was 50%. Eight parallel samples were prepared for the texture analysis. 

### 2.7. Analysis of Flavor Characteristics of Zhayu

#### 2.7.1. Sensory Analysis

Sensory evaluation of the taste and odor of Zhayu samples was performed by eight experienced panelists (four males and four females, aged 22 to 35) in a sensory laboratory at room temperature (22 ± 2 °C). For sensory evaluation, Zhayu samples were marked randomly with a three-digit number. Before tasting each sample, the panelists were required to rinse their mouths thoroughly with purified water, and the procedure of sensory evaluation was conducted following ISO 4120 [26] and ISO 4121 [27]. In this research, the intensities of sourness, bitterness, sweetness, umami, and saltness were evaluated as taste characteristics. Five attributes that best expressed fermented fish products’ aroma characteristics were selected as “fresh”, “fishy”, “earthy”, “oily”, and “acidic”. Each characteristic was rated on the following scale: 0, no taste/odor; 1, very weak; 2, weak; 3, moderate; 4, strong; and 5, very strong. The Zhayu prepared by natural fermentation was set as a reference. The standard scores of the reference were determined as sourness, 3.5; bitterness, 2; sweetness, 2; umami, 3; saltness, 3; fresh, 3.5; fishy, 1; earthy, 2; oily, 2.5; and acidic, 3.

#### 2.7.2. Electronic Tongue (E-Tongue) Analysis

E-tongue (ALPHA MOS, Heracles, France) analysis was performed to analyze the taste characteristics of Zhayu samples. The interaction-sensitive sensor system of E-tongue includes 7 chemical sensor arrays (AHS, PKS, CTS, NMS, CPS, ANS, and SCS) and an Ag/AgCl reference electrode. Each sensor shows various sensitivities of different tastes, and AHS, CTS, NMS, ANS, and SCS can reflect the intensity of sourness, saltness, umami, sweetness, and bitterness, respectively. A total of 20 g of Zhayu samples was homogenized with 100 mL of distilled water at 8000 r/min for 0.5 min. After centrifuging at 10,000 r/min for 10 min at 4 °C, 80 mL of the supernatant was used for the E-tongue analysis. E-tongue test conditions were set as follows: sample delay time, 0 s; acquisition time, 120 s; acquisition cycle, 1.00 s; stirring speed, 60 r/min [28]. Six parallel samples were prepared for the E-tongue analysis. 

#### 2.7.3. Electronic Nose (E-nose) Analysis

An E-nose (ALPHA MOS, Heracles, France) consists of an automatic sampling device, interaction-sensitive sensor array, data acquisition system, and data analysis software. To prepare samples for E-nose analysis, samples were cut into small pieces and 2 g of each sample was added to 10 mL headspace vials. The detection conditions of E-nose were set as follows: headspace inlet temperature, 50 °C; headspace time, 120 s; stirring speed, 500 r/min; injection volume, 2.5 mL; sample collection time, 120 s; delay time, 300 s [28]. Six parallel samples were prepared for the E-nose analysis.

#### 2.7.4. Determination of Free Amino Acids

Free amino acids in Zhayu samples (1 g) were extracted with 30 mL of hydrochloric acid (0.1 mol/L) by homogenizing at 6000 r/min for 3 min, and then shaking at ambient temperature for 15 min. After standing for 5 min, the supernatant was collected and the residue was washed twice with 20 mL hydrochloric acid (0.1 mol/L). All the obtained supernatants were pooled and brought to 100 mL with 0.1 mol/L hydrochloric acids. The free amino acid extract was filtrated through a 0.22 μm membrane, and 10 μL filtrate of each sample was taken for a derivative reaction. 

For the derivative reaction, the thermostat was first heated to 55 °C, followed by adding 70 μL of borate buffer in the AccQ·Tag Ultra Derivatization kit (Waters Corporation, Milford, MA, USA) and 10 μL of each sample solution to the sample vial successively, then vortex mixing immediately, and finally adding 20 μL of reconstituted AccQ·Tag Ultra reagent. After vortex mixing for 10 s and standing at room temperature for 1 min, the mixture was heated for 10 min at 55 °C and then analyzed by UHPLC (Dionex Ultimate 3000, Thermo Scientific, Sunnyvale, CA, USA). 

The amino acids were separated in an ACQUITY UPLC^®^ BEH C18 column (1.7 µm, 2.1 × 100 mm, Waters Corporation, Massachusetts, Ireland, USA) with the column temperature and detection wavelength at 55 °C and 260 nm, respectively. Eluent A was the concentrated solution in an AccQ·Tag Ultra reagent package (diluted 20 times with high-purity water), and eluent B was 100% acetonitrile. The elution procedure was initially 0.7 mL/min (A:B = 99.9:0.1), then 0.7 mL/min for 0.54 min (A:B = 99.9:0.1), 0.7 mL/min for 5.20 min (A:B = 90.9:9.1), 0.7 mL/min for 2 min (A:B = 10:90), 0.7 mL/min for 0.3 min (A:B = 40.4:59.6), 0.7 mL/min for 0.01 min (A:B = 10:90), 0.7 mL/min for 0.59 min (A:B = 10:90), 0.7 mL/min for 0.09 min (A:B = 99.9:0.1), and 0.7 mL/min for 0.77 min (A:B = 99.9:0.1) [28]. Amino acid standards were supplied from Waters Corporation and were derivatized and separated under the same conditions as the samples. The external standard curves were built to calculate the concentrations of amino acids in the samples. The chromatogram of the standards of amino acids is shown in Figure 2. 

#### 2.7.5. GC-MS Analysis

Volatile aroma compounds in Zhayu were identified and semi-quantitated by headspace–solid-phase microextraction (HS-SPME) and GC-MS. The GC-MS analysis was performed on an Agilent 7890B GC equipped with an Agilent 5977B mass selective detector (Agilent Technologies, Inc., Santa Clara, CA, USA). Chopped Zhayu samples (4.0 g) were added to 6 mL of saturated sodium chloride solution in a 20 mL autosampler glass. After a volume of 10 μL of an internal standard (50 μg/mL 2-octanol) was added, the autosampler glass was capped tightly with a Teflon-faced silicone septum. The samples were equilibrated at 45 °C for 15 min, and a DVB/CAR/PDMS fiber (50/30 μm, Supelco, Inc., Bellefonte, PA, USA) was used to extract volatiles for 40 min from headspace. The stirring speed was 400 r/min. The volatiles were desorbed into the GC injection port in a splitless mode at 250 °C. The desorption time was 5 min. The separation was performed using a DB-wax column (30 m length, 0.25 mm i.d., 0.25 μm film thickness; Agilent Technologies, Inc., Santa Clara, CA, USA). The oven temperature was programmed as follows: 40 °C (initial hold for 4 min), ramp at 4 °C/min to 230 °C (hold for 5 min). The carrier gas was nitrogen with a constant flow rate of 1.0 mL/min. The electron impact (EI) energy was 70 eV, and the ion source temperature was set at 230 °C. Each sample was run in triplicate. Mass spectra of compounds were compared to those in the National Institute of Standards and Technology (NIST) library (Agilent Technologies, Inc., Santa Clara, CA, USA). A standard mixture of n-alkanes (C6~C26) was prepared and injected into GC using the same conditions as the samples. Retention indices (RIs) were calculated following a modified Kováts method [29]. Compounds were positively identified by comparing mass spectra and RIs of the standards obtained in the laboratory or tentatively identified if the RIs were from the literature. The relative contents of volatiles were expressed by the ratio of the peak area of the volatiles to that of the internal standard.

### 2.8. Statistical Analysis

All tests were performed in triplicate. Results of E-nose and E-tongue data were processed through Alpha Soft 12.3 software (ALPHA MOS, Heracles, France). Variance analysis and principal component analysis (PCA) were conducted using SPSS 22.0 software (SPSS Inc., Chicago, IL, USA). Differences among mean values were established using Duncan’s multiple range test. *p* < 0.05 was considered statistically significant.

## 3. Results and Discussions

### 3.1. Safety Qualities

The pH values and TA contents are important indexes to evaluate the safety of fermented foods, and it was generally believed that low pH value (<4.5) and high TA content could inhibit the growth of spoilage bacteria and ensure the safety of products [30]. Figure 3 illustrates that the spontaneously fermented Zhayu samples showed the highest pH value at 5.15 and the lowest TA content at 13.41 mg/kg. The fermentation with LAB decreased pH values and increased TA content in Zhayu products significantly (*p* < 0.05). As the inoculation amount increased, the pH value decreased, and the TA content increased gradually, indicating the accumulation of LAB metabolites and the enhancement of the acidity of fermented products. When 10^9^ cfu/100 g of *P. acidilactici* and *P. pentosaceus* were inoculated, or the inoculation amounts of *L. plantarum* exceeded 10^7^ cfu/100 g, pH values of the fermented Zhayu were below 4.4, which is safe for consumption [31]. Additionally, samples fermented with *P. acidilactici* showed the lowest TA content compared with the other two starter cultures, indicating the differences in the acid production capacity of various LAB species. In other fermented fish samples, such as fermented silver carp sausage and plaa-som, pH values also decreased during the fermentation process with LAB, and the decrease in pH values was related to the growth of LAB [2,32]. 

The TVB-N content was closely related to the content of N-containing compounds (such as biological amines) produced by the decarboxylation of amino acids and the propagation of spoilage bacteria during the fermentation process, which could measure the spoilage of fermented fish products [33]. The contents of TVB-N of all Zhayu samples (21.0~30.8 mg/100 g) were lower than the superior limit provided by the European Union (≤35 mg/100 g), indicating that the edible safety qualities of the products were guaranteed (Figure 3c). Zhayu samples prepared with LAB showed significantly lower TVB-N contents compared with the control (*p* < 0.05); however, the concentrations of *L. plantarum* and *P. acidilactici* showed no significant effect on the TVB-N contents in Zhayu samples (*p* > 0.05), and the Zhayu samples cultured with *P. pentosaceus* (10^7^~10^9^ cfu/100 g) presented the lowest TVB-N contents. It was reported that the addition of LAB could increase the acid-producing metabolism, resulting in the decrease in pH value and the formation of some bacteriostatic substances which could inhibit the growth of spoilage bacteria such as *Enterobacter*, and then reduce the accumulation of TVB-N [34,35]. Additionally, in Zhayu samples prepared with LAB, the increased acids could neutralize alkaline ammonia and amines, reducing the TVB-N contents. Among the three kinds of LAB, *P. acidilactici* presented a weak ability to improve the safety qualities of Zhayu samples due to its poor acid production capacity, while *P. pentosaceus* could enhance the safety qualities of Zhayu.

### 3.2. Nutritional Properties

The crude fat, water-soluble proteins, and soluble solids were used to evaluate the nutritional properties of Zhayu samples. The contents of these are illustrated in Figure 3d–f. It was shown that the effects of LAB-inoculated fermentation on the nutritional properties of Zhayu depended on the LAB species and concentrations. When the inoculation amounts were 10^6^ cfu/100 g and 10^7^ cfu/100 g, the addition of LAB did not significantly affect the contents of crude fat in the Zhayu products (*p* > 0.05). However, the contents of crude fat noticeably increased in the samples fermented with 10^9^ cfu/100 g of *L. plantarum* and *P. acidilactici*, or with more than 10^8^ cfu/100g of *P. pentosaceus*. LAB could produce intracellular lipase and extracellular lipase during fermentation, which could promote the degradation of lipids and increase the content of crude fat in fermented products [36]. However, the lipolysis ability of LAB was relatively weak [37]. Thus, only high amounts of LAB addition could affect the contents of crude fat in fermented fish products. 

During fermentation, proteins in fish products were hydrolyzed by endogenous and microbial proteolytic enzymes to form more water-soluble proteins with low molecular weight, which were easily digested and absorbed [5]. However, due to the weak proteolytic activity of LAB, proteins in the LAB-inoculated Zhayu samples could be mainly hydrolyzed by endogenous enzymes such as cathepsins [15,37]. When the inoculation amount increased over 10^8^ cfu/100 g for *P. acidilactici* and *P. pentosaceus,* and the amount increased to 10^9^ cfu/100g for *L. plantarum*, the water-soluble protein content in Zhayu products noticeably increased (*p* < 0.05) compared with the control. Additionally, the content of water-soluble proteins was the highest in the sample prepared with 10^9^ cfu/100 g *P. pentosaceus*. In other fermented products, it was also found that starter culture showed a pronounced effect on protein degradation [7,32]. This suggests that the decrease in the pH value after fermentation with LAB promotes the activity of endogenous protease and then increases the content of water-soluble proteins [15].

The soluble solids in fermented fish products mainly include soluble sugar, vitamins, and minerals. Figure 3f shows that the addition of *L. plantarum* and *P. acidilactici* increased the content of soluble solids in fermented Zhayu products, while the soluble solids content of the samples fermented with *P. pentosaceus* showed a decrease compared with the control. During fermentation, the metabolism of microorganisms promotes the dissolution of soluble solids in cells, but the metabolism also consumes the soluble solids.

### 3.3. Texture Properties

The textural properties of fermented Zhayu products with and without starter cultures are shown in Table 1. The addition of *L. plantarum* and *P. acidilactici* significantly decreased the hardness of Zhayu samples (*p* < 0.05), but the amounts of *L. plantarum* and *P. acidilactici* did not significantly affect the hardness of the samples. When the inoculation concentration of LAB was 10^9^ cfu/100 g, samples that were fermented with *P. pentosaceus* showed the lowest hardness. The reason for the decrease in hardness was the degradation of proteins caused by endogenous enzymes, resulting in a decrease in the interactions between fibrillin and the combination of musculature. The trend of the hardness of Zhayu was consistent with the water-soluble proteins’ results (Figure 3e). However, it was found that fermentation with starter cultures could enhance the hardness of fermented fish sausages, Som-fug, and other fermented products [15,34,38]. This might be due to the different manufacturing processes of these fermented products. Fermentation with LAB starters also decreased the springiness and the chewiness of Zhayu products. Samples fermented with *P. pentosaceus* showed higher springiness and chewiness. It was suggested that Zhayu samples fermented with 10^9^ cfu/100 g *P. pentosaceus* were tender, softer, and elastic, and presented better textural properties that were acceptable for consumers compared with other samples. 

### 3.4. Taste Characteristics

The above results show that Zhayu samples fermented with each LAB at the inoculation amount of 10^9^ cfu/100 g presented high safety qualities, contained high fats and water-soluble proteins, and showed tender and soft texture properties. Therefore, the Zhayu samples inoculated with 10^9^ cfu/100 g of each LAB were chosen as the typical samples to compare the effects of LAB species on the flavor characteristics of Zhayu samples. Principal component analysis (PCA) was performed based on the responses of the E-tongue chemical sensor arrays for different Zhayu products to compare their taste profiles. Figure 4a illustrates the PCA results, and the first two principal components described 99.32% of the total taste variances, indicating that the taste characteristics of Zhayu products fermented with three kinds of LAB and the control were various. It was noticeable that the samples fermented with *L. plantarum* and *P. acidilactici* showed more similar taste characteristics. 

A sensory evaluation was carried out to study the effects of LAB on the tastes of Zhayu products (Figure 4b). The inoculation of LAB dramatically increased the sourness, saltness, and umami taste of the Zhayu samples. The samples with the addition of *L. plantarum* and *P. acidilactici* presented a relatively strong sourness, while the samples fermented with *P. pentosaceus* showed the highest level of umami taste. The increase in sensory sourness was coincident with the rise in TA in Zhayu samples after starter culture fermentation (Figure 3), due to the lactic acid produced by LAB metabolism and the acid denaturation of proteins [34]. The umami taste in fish-related products mainly originates from the catabolism of proteins and nucleotides [39]. It was suggested that the positive effect of *P. pentosaceus* on the umami taste might be related to the degradation of proteins during fermentation. Meanwhile, the increase in umami tastes promoted the perception of saltiness in Zhayu samples, because of the interactions between the flavor substances [40]. Additionally, compared with the control, the inoculation with *P. pentosaceus* increased the sweetness and bitterness of Zhayu samples, while the addition of *L. plantarum* and *P. acidilactici* reduced the sweet and bitter perceptions. These differences might be due to the different protein degradation results of each LAB.

During the fermentation of aquatic products, proteins were hydrolyzed by microbes and endogenous protease, promoting the formation of free amino acids and changes in taste properties [7,41]. In Zhayu samples, the addition of LAB increased most of the contents of free amino acids (Table 2). In particular, Asp and Glu, which have an umami taste and relatively low thresholds, presented significantly higher contents and taste activity values (TAVs) in LAB-inoculated samples than those in the control. Moreover, Zhayu samples fermented with *P. pentosaceus* contained the most amounts of Asp and Glu, which was consistent with the highest water-soluble protein contents (Figure 3e) and the highest sensory umami scores (Figure 4b). In fish sauce and fish chili paste, it was also found that the inoculation fermentation significantly increased the content of free amino acids with the umami taste (Glu and Asp) [14,42]. Additionally, the contents of Thr, Lys, Tyr, and His, with a sweet or bitter taste, were higher in the LAB-inoculated Zhayu samples than in the control, suggesting that fermentation with LAB could enhance the hierarchy of taste perception of Zhayu samples.

### 3.5. Odor Characteristics

Figure 4c,d illustrates the PCA results based on the responses of the E-nose sensors and the sensory results for the odor characteristics of different Zhayu samples. The first two principal components described 97.06% of the total odor variances, indicating that the responses of E-nose sensors could adequately differentiate Zhayu products fermented with various LAB. The PCA results indicate that the odor properties of each fermented Zhayu sample differed significantly. Additionally, Zhayu samples fermented with LAB showed stronger fresh and acidic notes, but weaker fishy, earthy, and oily odors compared with the control (Figure 4d), suggesting that the LAB-inoculated fermentation could enhance the aroma characteristics of Zhayu samples. Especially for the samples inoculated with *P. acidilactici*, the promotion of the pleasant fresh odor and the inhibition of earthy notes were the most pronounced. Moreover, samples prepared with *L. plantarum* presented the strongest acidic odor. 

To further understand the differences in volatile compounds of Zhayu samples prepared by spontaneous and inoculated fermentation, the relative concentrations of volatile compounds in Zhayu samples were detected by HS-SPME-C-MS. The results of the relative contents of volatile compounds in the typical Zhayu samples are shown in Table 3. 

Aldehydes, such as hexanal with a grassy note, octanal with a citrus odor, and (E,E)-2,4-decadienal with a fishy or fatty smell, are reported as major aroma compounds in fish-related products due to their low odor thresholds, and they are mainly derived from the oxidation of fatty acids and the degradation of proteins [8,9,38]. It was noticeable that except nonanal and benzaldehyde, the relative contents of most aldehydes were significantly lower in Zhayu samples fermented with LAB than those in the control. Especially, the amount of hexanal in the control was approximately 5.1~5.5 times higher than that in the LAB-inoculated fermented Zhayu samples. Additionally, octanal and nonanal were not detected in the Zhayu samples inoculated with *P. pentosaceus.* It was reported that in other fermented fish samples (such as *Suan zuo yu* and fermented mandarin fish), aldehydes exhibited a downward trend during fermentation [8,9]. Moreover, starter cultures showed an antioxidant capacity on unsaturated fatty acids in fermented fish products, reducing the concentrations of aliphatic aldehydes [13,42]. Additionally, it was reported that in fish paste, *L. plantarum* significantly reduced the lipolysis and inhibited the flavor generation by the other strains [14]. Thus, it was presumed that the inoculated fermentation with LAB might inhibit the lipid oxidation in Zhayu samples, decreasing the aldehydes with grassy, fishy, or fatty notes, and then increasing the aroma properties of Zhayu. 

Fourteen kinds of alcohols were identified in Zhayu samples. However, most of them showed a high odor threshold and contributed little to the aroma profiles of fish-related products, except (Z)-3-hexen-1-ol, 1-octen-3-ol, and 1-heptanol [6,8]. Among them, 1-penten-3-ol, (Z)-3-hexen-1-ol, and 1-octen-3-ol, which are mainly derived from the oxidation of unsaturated fatty acids and have grassy, fishy, and mushroom-like notes [8], presented the lowest contents in Zhayu samples fermented with *P. acidilactici*, consistent with the weakest notes of “earthy” and “fishy” by the sensory analysis (Figure 4d). However, the LAB-inoculated fermentation increased the concentrations of 1-heptanol, and so did other saturated alcohols in the samples. This might be because the role of microorganisms also had an important influence on the aroma property of fermented aquatic products, which could increase the content of alcohols [16]. 

Zhayu samples contained many terpenoids. The source of the terpenoids in fish-related products was related to the addition of hot pepper, ginger, and other seasonings [45]. Interestingly, the relative contents of many terpenoids, contributing to lemon-like, woody, or floral notes, were noticeably higher in the samples fermented with LAB than in the samples prepared by spontaneous fermentation. It was suggested that during the fermentation of Zhayu, the metabolism of LAB could promote the formation of terpenoids. A similar phenomenon was also found in Suan yu, another kind of fermented fish product [6]. The enriched terpenoids by LAB-inoculated fermentation would enhance the overall aroma characteristics of the fermented fish products. 

For other compounds, the contents of ketones in LAB-inoculated Zhayu samples were significantly lower than those in the control, especially 2,3-butanedione with a buttery note and 3-hydroxy-2-butanone with a yogurt-like note. Not surprisingly, the LAB-inoculated fermentation dramatically increased the acids’ contents in the Zhayu samples compared with the control, which was coincident with the sensory evaluation results (Figure 4d). Additionally, the Zhayu samples prepared with *L. plantarum* contained the highest acetic acid, butanoic acid, and 4-methylpentanoic acid. Meanwhile, esters, which were derived from the microbial esterification of acids with alcohols, also showed higher contents in Zhayu samples fermented with LAB than in the control. Esters usually had a fruity aroma, and they are important aroma compounds contributing to the unique odor of fermented products, such as dry-fermented sausages and fermented mandarin fish [8,46].

It was found that the inoculated fermentation with LAB promoted the generation of the compounds that derived from protein degradation, increasing the relative content of aromatic compounds (such as benzyl alcohol, phenylethyl alcohol, and methyl salicylate), phenols (except eugenol), and S-containing compounds. In particular, dimethyl sulfide with a garlic-like odor showed dramatically higher relative contents (45.39~70.47 μg/kg) in the Zhayu samples fermented with LAB than in the control (5.71 μg/kg). Moreover, the samples fermented with *L. plantarum* and *P. pentosaceus* contained more dimethyl sulfide and dimethyl disulfide. S-containing compounds were produced by the degradation of S-containing amino acids and the catabolism of aromatic [16], and they were also identified as major contributors to the typical fishy odor of fermented fish products [8]. The contents of Met in the LAB-inoculated Zhayu samples were also lower than those in the control (Table 2), suggesting that LAB could utilize Met to produce volatile S-containing compounds during fermentation [8,16]. However, some N-containing compounds, which were also generated by the protein degradation, decreased after fermentation with LAB (especially *L. plantarum* and *P. pentosaceus*), agreeing with the changes in TVB-N (Figure 3c). It was reported that N-containing compounds could be gradually consumed during the fermentation process of aquatic products [9]. Thus, it was presumed that the decrease in the N-containing compounds in the LAB-inoculated Zhayu samples was due to the further microbial decomposition and utilization of these compounds by microorganisms. 

## 4. Conclusions

The Zhayu products fermented with LAB had lower pH values and TVB-N contents than the control, showing better safety qualities. Moreover, LAB-inoculated Zhayu samples contained more water-soluble proteins and crude fats as the inoculation amount exceeded 10^8^ cfu/100 g, and fermentation with LAB decreased the hardness, springiness, and chewiness of Zhayu products, making the products more tender and soft. Additionally, the Zhayu prepared with LAB at the inoculation amount of 10^9^ cfu/100 g presented better safety, nutritional, and textural qualities. LAB could also improve the flavor characteristics of Zhayu products. Zhayu samples fermented with *L. plantarum* and *P. acidilactici* presented the strongest sourness, while Zhayu samples prepared with *P. pentosaceus* showed the highest level of umami taste, consistent with the highest contents of water-soluble proteins and free amino acids with an umami taste (Asp and Glu). For odor characteristics, the addition of LAB decreased the relative contents of aliphatic aldehydes, (Z)-3-hexen-1-ol, and 1-octen-3-ol, subducting the earthy and fishy notes in the Zhayu samples, and the Zhayu samples fermented with *P. acidilactici* contained the lowest content of 1-octen-3-ol and the weakest earthy notes. In addition, LAB-inoculated fermentation enhanced the contents of terpenoids, acids, esters, and S-containing compounds, increasing the sour, pleasant, and unique odors of the Zhayu products. Especially for samples fermented with *L. plantarum*, the relative contents of acids and esters were the highest. Overall, the inoculation fermentation with LAB increased the acceptability of the comprehensive quality of Zhayu products. In further studies, the three kinds of LAB can be mixed and used together to exploit the advantages of each starter and improve the quality of fermented fish products.

## Figures and Tables

**Figure 1 foods-11-02756-f001:**
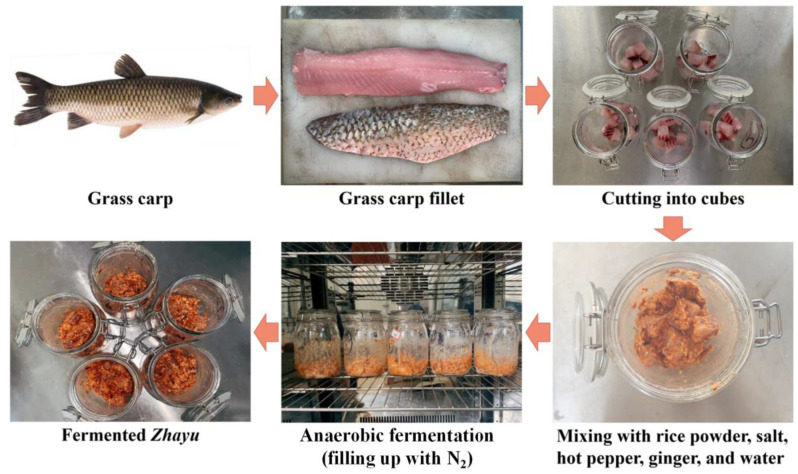
Schematic illustration for the process of fermented Zhayu products.

**Figure 2 foods-11-02756-f002:**
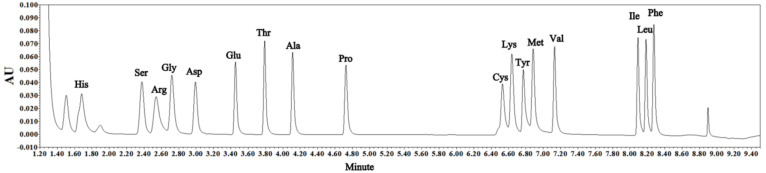
Chromatogram of the standards of amino acids detected by UHPLC.

**Figure 3 foods-11-02756-f003:**
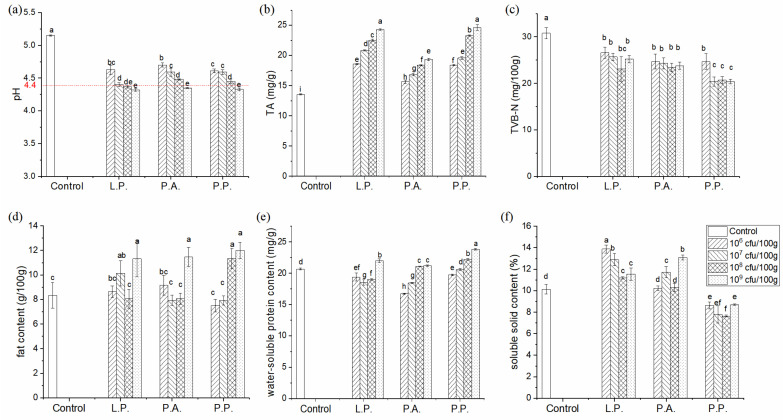
The pH values (**a**), TA contents (**b**), TVB-N contents (**c**), crude fat contents (**d**), water-soluble protein contents (**e**), and soluble solid contents (**f**) of Zhayu products prepared by spontaneous fermentation (control) and fermented with *L. plantarum* (L.P.), *P. acidilactici* (P.A.), and *P. pentosaceus* (P.P.) at different inoculation amounts. Different letters on the bar denote significant differences (*p* < 0.05).

**Figure 4 foods-11-02756-f004:**
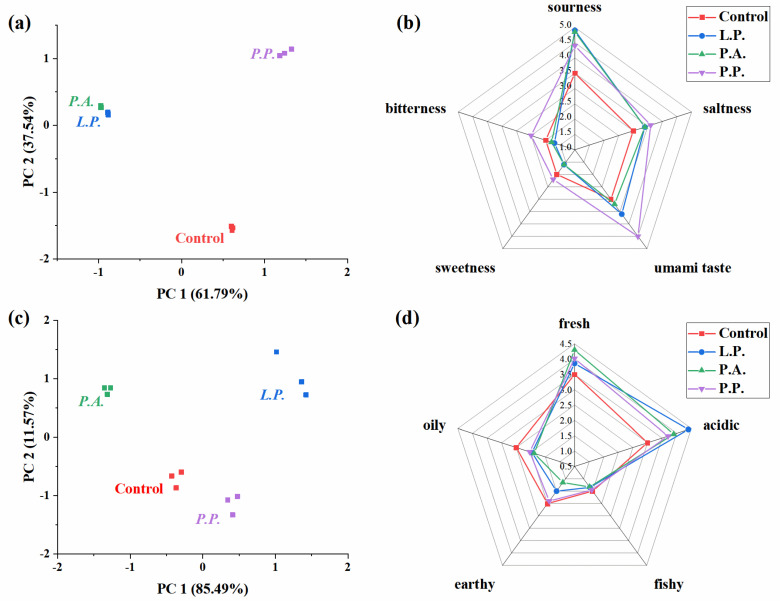
Principal component analysis (PCA) of Zhayu samples fermented with different starter cultures based on E-tongue results (**a**) and E-nose results (**c**), and sensory evaluation of taste characteristics (**b**) and odor profiles (**d**) of Zhayu products prepared with different kinds of LAB.

**Table 1 foods-11-02756-t001:** Textural properties of Zhayu samples prepared by spontaneous fermentation (Control) and different inoculation amounts of *L. plantarum*, *P. acidilactici*, and *P. pentosaceus*.

Inoculation Amounts (cfu/100 g)	Hardness (g)	Springiness	Chewiness (g)
Control	250.16 ± 31.54 ^a^	1.00 ± 0.01 ^a^	163.4 ± 20.3 ^a^
*L. plantarum*			
10^6^	203.02 ± 26.84 ^bcd^	0.71 ± 0.06 ^ef^	45.53 ± 18.11 ^ef^
10^7^	171.64 ± 16.33 ^de^	0.80 ± 0.19 ^ef^	42.86 ± 9.95 ^ef^
10^8^	176.85 ± 21.91 ^cde^	0.88 ± 0.09 ^de^	54.19 ± 9.04 ^de^
10^9^	186.78±20.6 ^cd^	0.75 ± 0.10 ^f^	31.66 ± 9.43 ^f^
*P. acidilactici*			
10^6^	178.21 ± 3.90 ^cd^	0.99 ± 0.01 ^ab^	111.4 ± 2.8 ^bc^
10^7^	171.81 ± 32.46 ^de^	0.97 ± 0.02 ^b^	61.39 ± 2.65 ^de^
10^8^	174.32 ± 26.46 ^cde^	0.79 ± 0.09 ^d^	67.31 ± 5.12 ^d^
10^9^	162.28 ± 16.33 ^de^	0.84 ± 0.09 ^ef^	44.31 ± 6.26 ^ef^
*P. pentosaceus*			
10^6^	246.96 ± 26.27 ^ab^	0.87 ± 0.06 ^cd^	125.1 ± 3.6 ^b^
10^7^	247.29 ± 35.79 ^ab^	0.84 ± 0.03 ^d^	94.07 ± 4.92 ^c^
10^8^	220.27 ± 11.41 ^abc^	0.91 ± 0.03 ^c^	98.97 ± 19.33 ^c^
10^9^	130.35 ± 31.86 ^e^	0.94 ± 0.06 ^bc^	109.6 ± 18.2 ^bc^

Notes: Different letters denote significant differences in the same column (*p* < 0.05).

**Table 2 foods-11-02756-t002:** Contents (mg/100 g) and taste activity value (TAV) of free amino acids in Zhayu samples prepared by spontaneous fermentation (SF) and fermented with 109 cfu/100 g of *L. plantarum* (L.P.), *P. acidilactici* (P.A.), and *P. pentosaceus* (P.P.).

FAA	Threshold (mg/100 g)	Taste	Concentration (mg/100 g)				TAV			
			SF	L.P.	P.A.	P.P.	SF	L.P.	P.A.	P.P.
Asp	3	umami	5.51 ± 0.32 ^d^	45.6 ± 1.7 ^b^	42.1 ± 1.1 ^c^	52.1 ± 1.9 ^a^	1.84	15.19	14.05	17.36
Glu	5	umami	49.0 ± 3.3 ^d^	56.8 ± 2.7 ^b^	51.8 ± 1.6 ^c^	67.8 ± 2.47 ^a^	9.80	14.05	17.26	22.61
Ser	150	sweet	4.61 ± 0.26 ^a^	2.97 ± 0.31 ^c^	2.24 ± 0.06 ^d^	3.66 ± 0.13 ^b^	0.03	0.02	0.01	0.02
Gly	150	sweet	23.2 ± 1.3 ^a^	19.4 ± 0.8 ^c^	21.1 ± 0.5 ^b^	21.6 ± 0.7 ^bc^	0.15	0.13	0.14	0.14
Thr	260	sweet	9.26 ± 0.53 ^c^	14.2 ± 0.4 ^b^	17.8 ± 0.4 ^a^	18.4 ± 0.7 ^a^	0.04	0.05	0.07	0.07
Ala	60	sweet	58.5 ± 3.3 ^a^	44.3 ± 1.4 ^c^	45.2 ± 1.2 ^c^	47.5 ± 1.7 ^b^	0.97	0.74	0.75	0.79
Pro	300	sweet/bitter	51.9 ± 2.9 ^b^	51.8 ± 3.5 ^b^	46.5 ± 1.2 ^c^	56.6 ± 2.1 ^a^	0.17	0.17	0.15	0.19
Lys	50	sweet/bitter	12.6 ± 0.7 ^c^	45.1 ± 1.2 ^ab^	44.2 ± 1.1 ^b^	46.4 ± 1.6 ^a^	0.25	0.90	0.88	0.93
Arg	50	bitter	1.97 ± 0.11 ^b^	2.53 ± 0.36 ^a^	1.11 ± 0.03 ^d^	1.48 ± 0.05 ^c^	0.04	0.05	0.02	0.03
Val	40	bitter	30.3 ± 1.7 ^a^	25.5 ± 0.7 ^c^	26.2 ± 0.7 ^bc^	27.0 ± 0.9 ^b^	0.76	0.64	0.65	0.68
Tyr	N.A.	bitter	20.8 ± 1.2 ^c^	27.7 ± 0.7 ^a^	24.6 ± 0.6 ^b^	25.9 ± 0.9 ^b^				
His	20	bitter	30.4 ± 1.7 ^c^	76.4 ± 3.6 ^a^	79.8 ± 2.1 ^a^	70.8 ± 2.5 ^b^	1.52	3.82	3.99	3.54
Leu	190	bitter	50.5 ± 2.9 ^b^	53.4 ± 1.6 ^a^	50.9 ± 1.3 ^b^	53.5 ± 1.9 ^a^	0.27	0.28	0.27	0.28
Ile	90	bitter	23.2 ± 1.3 ^a^	18.1 ± 0.7 ^b^	17.5 ± 0.5 ^b^	18.4 ± 0.7 ^b^	0.26	0.20	0.19	0.20
Phe	90	bitter	38.2 ± 2.1 ^ab^	41.1 ± 1.0 ^a^	37.7 ± 0.9 ^b^	39.0 ± 1.4 ^ab^	0.42	0.46	0.42	0.43
Met	30	bitter	17.8 ± 1.0 ^a^	15.2 ± 0.6 ^b^	14.3 ± 0.3 ^c^	14.9 ± 0.5 ^bc^	0.59	0.51	0.48	0.50
Cys	N.A.	salt	N.D.	N.D.	N.D.	N.D.				

Notes: N.A., not available; N.D., not detected. Different letters denote significant differences in the same row (*p* < 0.05). Thresholds for amino acids adapted from Wang et al. [38].

**Table 3 foods-11-02756-t003:** Relative contents (μg/kg) of aroma compounds in Zhayu samples prepared by spontaneous fermentation (SF) and fermented with 10^9^ cfu/100 g of *L. plantarum* (L.P.), *P. acidilactici* (P.A.), and *P. pentosaceus* (P.P.).

Compounds	RI	Identification	Odor	Odor Threshold (μg/kg)	Relative Concentration (μg/kg)			
					SF	L.P.	P.A.	P.P.
Aldehydes								
2/3-methylbutanal	923	MS, RIL	malty	1.5/0.5 ^b^	0.74 ± 0.13 ^a^	0.51 ± 0.05 ^c^	0.61 ± 0.04 ^b^	0.67 ± 0.07 ^b^
hexanal	1087	MS, RI	grassy	2.4 ^a^	2.52 ± 0.78 ^a^	0.48 ± 0.02 ^b^	0.49 ± 0.09 ^b^	0.46 ± 0.17 ^b^
octanal	1294	MS, RI	citrus	3.4 ^a^	0.28 ± 0.13 ^a^	0.11 ± 0.03 ^c^	0.15 ± 0.04 ^b^	N.D.
(E)-2-heptenal	1326	MS, RI	green	13 ^c^	0.48 ± 0.07 ^a^	0.41 ± 0.05 ^b^	0.45 ± 0.06 ^ab^	0.40 ± 0.04 ^b^
nonanal	1399	MS, RI	citrus	2.8 ^a^	0.09 ± 0.02 ^a^	0.11 ± 0.02 ^a^	0.12 ± 0.03 ^a^	N.D.
(E)-2-octenal	1435	MS, RI	nutty	3 ^a^	1.17 ± 0.03 ^a^	0.59 ± 0.05 ^b^	0.61 ± 0.09 ^b^	0.66 ± 0.07 ^b^
benzaldehyde	1508	MS, RI	rosy	350–3500 ^c^	1.35 ± 0.36 ^a^	1.07 ± 0.00 ^a^	1.17 ± 0.03 ^a^	1.02 ± 0.19 ^a^
(E,E)-2,4-decadienal	1820	MS, RI	fishy, fatty	0.027 ^a^	1.64 ± 0.11 ^a^	0.85 ± 0.05 ^b^	0.86 ± 0.06 ^b^	0.67 ± 0.12 ^c^
Alcohols								
1-propanol	1049	MS, RIL	alcoholic	7000 ^c^	0.29 ± 0.02 ^c^	0.38 ± 0.16 ^b^	0.54 ± 0.00 ^b^	0.89 ± 0.05 ^a^
1-penten-3-ol	1164	MS, RI	grassy	400 ^c^	0.76 ± 0.16 ^a^	0.56 ± 0.05 ^a^	0.42 ± 0.08 ^b^	0.65 ± 0.12 ^a^
1-pentanol	1260	MS, RIL	fusel-like	4000 ^c^	3.35 ± 0.88 ^a^	1.93 ± 0.14 ^b^	1.75 ± 0.25 ^b^	1.85 ± 0.15 ^b^
2-heptanol	1322	MS, RIL	herbaceous	41–81 ^c^	2.41 ± 0.35 ^b^	5.06 ± 0.02 ^a^	4.34 ± 0.21 ^a^	4.25 ± 0.83 ^a^
1-hexanol	1359	MS, RI	herbaceous	2500 ^c^	7.19 ± 0.84 ^ab^	8.14 ± 0.31 ^a^	6.00 ± 0.42 ^b^	8.29 ± 0.88 ^a^
(Z)-3-hexen-1-ol	1385	MS, RI	fishy	3.9 ^b^	1.86 ± 0.38 ^a^	1.57 ± 0.09 ^a^	1.27 ± 0.07 ^b^	1.74 ± 0.13 ^a^
3-octanol	1390	MS, RIL	oily, nutty	18–250 ^c^	0.46 ± 0.07 ^b^	0.67 ± 0.08 ^a^	0.67 ± 0.04 ^a^	0.62 ± 0.18 ^ab^
1-octen-3-ol	1457	MS, RI	mushroom	1.2 ^a^	1.64 ± 0.34 ^a^	1.49 ± 0.29 ^a^	0.51 ± 0.06 ^b^	1.27 ± 0.48 ^a^
1-heptanol	1468	MS, RIL	woody	3 ^c^	0.21 ± 0.03 ^c^	0.90 ± 0.07 ^a^	0.65 ± 0.06 ^b^	0.58 ± 0.07 ^b^
2-ethylhexanol	1488	MS, RIL	green, oily	270,000 ^c^	0.41 ± 0.07 ^b^	1.01 ± 0.13 ^a^	0.67 ± 0.12 ^ab^	0.98 ± 0.27 ^a^
1-octanol	1575	MS, RIL	soapy	110–130 ^c^	0.19 ± 0.03 ^c^	0.44 ± 0.01 ^a^	0.30 ± 0.01 ^b^	0.31 ± 0.09 ^b^
1-nonanol	1663	MS, RIL	soapy	50 ^c^	0.15 ± 0.00 ^c^	0.73 ± 0.09 ^a^	0.36 ± 0.03 ^b^	0.43 ± 0.13 ^b^
benzyl alcohol	1880	MS, RI	rosy	1.2–1000 ^c^	0.73 ± 0.19 ^b^	0.87 ± 0.11 ^ab^	0.92 ± 0.02 ^a^	1.05 ± 0.18 ^a^
phenylethyl alcohol	1922	MS, RI	rosy	140 ^b^	0.43 ± 0.01 ^b^	0.54 ± 0.05 ^a^	0.56 ± 0.03 ^a^	0.67 ± 0.14 ^a^
Terpenoids								
α-pinene	1036	MS, RIL	pine	2.2 ^b^	3.41 ± 1.33 ^a^	1.41 ± 0.16 ^b^	1.93 ± 0.23 ^a^	2.43 ± 0.34 ^a^
camphene	1059	MS, RIL	camphoraceous	N.A.	4.06 ± 0.99 ^b^	3.03 ± 0.21 ^b^	4.53 ± 1.79 ^b^	7.72 ± 1.12 ^a^
β-myrcene	1170	MS, RIL	balsamic	1.2 ^b^	0.56 ± 0.15 ^a^	0.15 ± 0.01 ^b^	0.63 ± 0.09 ^a^	0.47 ± 0.12 ^a^
D-limonene	1204	MS, RIL	lemon-like	10 ^c^	0.70 ± 0.27 ^b^	0.56 ± 0.07 ^c^	1.14 ± 0.01 ^a^	1.02 ± 0.23 ^ab^
β-phellandrene	1219	MS, RIL	citrus	40–200 ^c^	0.77 ± 0.07 ^b^	0.87 ± 0.04 ^b^	1.23 ± 0.70 ^a^	1.03 ± 0.28 ^a^
eucalyptol	1230	MS, RIL	camphoraceous	12 ^c^	12.46 ± 2.01 ^b^	14.26 ± 0.53 ^a^	15.15 ± 1.68 ^a^	15.35 ± 4.12 ^ab^
caryophyllene	1570	MS, RIL	woody	64 ^c^	N.D.	2.35 ± 0.58 ^a^	1.22 ± 0.34 ^b^	2.33 ± 0.14 ^a^
camphor	1524	MS, RIL	minty	1000–1290 ^c^	0.36 ± 0.07 ^c^	0.69 ± 0.05 ^b^	0.80 ± 0.01 ^a^	0.83 ± 0.18 ^ab^
linalool	1554	MS, RIL	lemon-like	0.87 ^b^	5.23 ± 0.70 ^b^	6.83 ± 0.26 ^a^	6.79 ± 0.22 ^a^	7.28 ± 1.70 ^a^
terpinen-4-ol	1635	MS, RIL	lilac-like	N.A.	0.26 ± 0.06 ^b^	0.58 ± 0.05 ^a^	0.56 ± 0.04 ^a^	0.57 ± 0.12 ^a^
citral	1663	MS, RIL	lemon-like	32 ^c^	0.09 ± 0.00 ^b^	0.58 ± 0.08 ^a^	0.68 ± 0.02 ^a^	0.53 ± 0.21 ^a^
α-terpineol	1692	MS, RIL	lilac-like	330 ^c^	0.55 ± 0.05 ^b^	0.88 ± 0.10 ^a^	0.90 ± 0.04 ^a^	1.05 ± 0.31 ^a^
borneol	1696	MS, RIL	piney	140 ^c^	1.62 ± 0.21 ^b^	3.24 ± 0.45 ^a^	3.28 ± 0.02 ^a^	4.11 ± 1.24 ^a^
β-bisabolene	1721	MS, RIL	balsamic	N.A.	0.25 ± 0.00 ^b^	0.73 ± 0.21 ^a^	0.66 ± 0.08 ^a^	1.23 ± 0.41 ^a^
neral	1733	MS, RIL	lemon-like	30 ^c^	0.38 ± 0.01 ^b^	0.59 ± 0.06 ^a^	0.73 ± 0.01 ^a^	0.64 ± 0.22 ^a^
nerol	1808	MS, RIL	rosy	300 ^c^	0.15 ± 0.03 ^c^	0.32 ± 0.00 ^a^	0.24 ± 0.01 ^b^	0.24 ± 0.07 ^b^
isogeraniol	1812	MS, RIL	rosy	N.A.	N.D.	1.23 ± 0.19 ^ab^	1.40 ± 0.08 ^a^	1.03 ± 0.13 ^b^
geraniol	1857	MS, RIL	rosy	1.1 ^b^	0.33 ± 0.04 ^c^	1.91 ± 0.15 ^a^	0.92 ± 0.10 ^b^	1.09 ± 0.41 ^b^
Ketones								
2,3-butanedione	968	MS, RI	buttery	1 ^b^	2.20 ± 0.93 ^a^	0.53 ± 0.02 ^b^	0.52 ± 0.03 ^b^	0.72 ± 0.17 ^b^
3-octanone	1240	MS, RIL	fruity	21–50 ^c^	0.27 ± 0.12 ^ab^	0.15 ± 0.00 ^b^	0.09 ± 0.01 ^c^	0.31 ± 0.04 ^a^
3-hydroxy-2-butanone	1286	MS, RI	yogurt	800 ^c^	15.11 ± 3.08 ^a^	1.87 ± 0.09 ^b^	1.56 ± 0.29 ^b^	1.55 ± 0.26 ^b^
6-methyl-5-hepten-2-one	1341	MS, RIL	green, fatty	50 ^c^	0.87 ± 0.14 ^a^	0.41 ± 0.04 ^b^	0.56 ± 0.10 ^b^	0.67 ± 0.22 ^ab^
Lactones								
butyrolactone	1634	MS, RI	sweet, buttery	20,000–50,000 ^c^	0.26 ± 0.04 ^a^	N.D.	N.D.	N.D.
lavender lactone	1684	MS, RIL	lavender	N.A.	0.48 ± 0.09 ^a^	0.43 ± 0.03 ^a^	0.44 ± 0.03 ^a^	0.50 ± 0.03 ^a^
γ-nonalactone	2042	MS, RI	coconut	9.7 ^b^	0.10 ± 0.01 ^c^	0.19 ± 0.01 ^b^	0.19 ± 0.02 ^b^	0.23 ± 0.01 ^a^
Acids								
acetic acid	1445	MS, RIL	acid	99,000 ^b^	6.93 ± 0.33 ^c^	17.73 ± 1.57 ^a^	13.53 ± 0.94 ^b^	13.28 ± 2.07 ^b^
butanoic acid	1576	MS, RI	rancid	2400 ^b^	0.12 ± 0.01 ^c^	0.41 ± 0.07 ^a^	0.36 ± 0.05 ^a^	0.26 ± 0.02 ^b^
2/3-methylbutanoic acid	1627	MS, RI	rancid	2200/490 ^b^	1.22 ± 0.21 ^a^	1.13 ± 0.07 ^a^	1.23 ± 0.24 ^a^	1.09 ± 0.10 ^a^
pentanoic acid	1720	MS, RIL	rancid	11,000 ^b^	0.10 ± 0.01 ^a^	0.08 ± 0.01 ^a^	0.09 ± 0.02 ^a^	0.07 ± 0.02 ^a^
4-methylpentanoic acid	1817	MS, RIL	rancid	810 ^c^	0.20 ± 0.02 ^b^	0.42 ± 0.08 ^a^	0.18 ± 0.01 ^b^	0.22 ± 0.04 ^b^
hexanoic acid	1839	MS, RI	rancid	890 ^a^	0.53 ± 0.14 ^a^	0.35 ± 0.04 ^b^	0.42 ± 0.06 ^a^	0.28 ± 0.09 ^b^
octanoic acid	2089	MS, RI	rancid	3000 ^a^	0.05 ± 0.01 ^c^	0.12 ± 0.01 ^b^	0.19 ± 0.04 ^a^	0.10 ± 0.01 ^b^
Esters								
ethyl acetate	880	MS, RI	fruity	5–5000 ^d^	0.86 ± 0.11 ^c^	1.12 ± 0.01 ^a^	1.07 ± 0.11 ^b^	0.92 ± 0.08 ^bc^
butyl acetate	1078	MS, RI	pineapple	66 ^c^	N.D.	0.04 ± 0.02 ^a^	0.03 ± 0.03 ^a^	N.D.
isoamyl acetate	1126	MS, RI	banana	2 ^d^	N.D.	0.07 ± 0.01 ^b^	0.13 ± 0.01 ^a^	N.D.
methyl salicylate	1759	MS, RIL	minty	40 ^d^	0.34 ± 0.07 ^b^	1.28 ± 0.05 ^a^	1.22 ± 0.04 ^a^	1.45 ± 0.32 ^a^
2-phenylethyl butanoate	1978	MS, RIL	rosy	N.A.	0.27 ± 0.04 ^ab^	0.29 ± 0.01 ^a^	0.22 ± 0.01 ^c^	0.25 ± 0.02 ^b^
Phenols								
2-methoxyphenol	1862	MS, RI	sweet	0.84 ^b^	4.49 ± 0.81 ^a^	0.35 ± 0.03 ^c^	0.36 ± 0.01 ^c^	0.53 ± 0.05 ^b^
phenol	2008	MS, RI	phenolic	5900 ^c^	1.63 ± 0.34 ^a^	0.28 ± 0.01 ^c^	0.29 ± 0.01 ^c^	0.64 ± 0.05 ^b^
2-methoxy-4-vinylphenol	2155	MS, RI	phenolic	5.1 ^b^	1.33 ± 0.19 ^a^	0.15 ± 0.01 ^c^	0.13 ± 0.02 ^c^	0.44 ± 0.09 ^b^
eugenol	2176	MS, RIL	clove	6–30 ^c^	2.22 ± 0.11 ^d^	8.29 ± 1.22 ^b^	6.32 ± 1.25 ^c^	11.23 ± 1.96 ^a^
S-containing compounds								
dimethyl sulfide	777	MS, RIL	garlic	0.84 ^a^	5.71 ± 1.73 ^c^	70.47 ± 6.99 ^a^	45.39 ± 3.73 ^b^	66.03 ± 11.74 ^a^
dimethyl disulfide	1107	MS, RIL	fishy	1.1 ^a^	N.D.	0.20 ± 0.07 ^a^	N.D.	0.09 ± 0.04 ^b^
N-containing compounds								
2-methylpyrazine	1258	MS, RIL	nutty, cocoa-like	60–100,000 ^c^	0.89 ± 0.17 ^a^	0.52 ± 0.04 ^d^	0.68 ± 0.03 ^b^	0.64 ± 0.01 ^c^
2,3-dimethylpyrazine	1339	MS, RIL	nutty, cocoa-like	2500–35,000 ^c^	0.37 ± 0.04 ^a^	0.13 ± 0.01 ^c^	0.24 ± 0.02 ^b^	0.11 ± 0.02 ^c^
2-ethyl-6-methylpyrazine	1377	MS, RIL	roasted	N.A.	0.46 ± 0.08 ^a^	0.28 ± 0.01 ^b^	0.48 ± 0.07 ^a^	0.29 ± 0.01 ^b^
trimethylpyrazine	1399	MS, RIL	baked potato	400–1800 ^c^	0.13 ± 0.01 ^a^	0.11 ± 0.01 ^b^	0.14 ± 0.02 ^a^	0.11 ± 0.01 ^b^
tetramethylpyrazine	1472	MS, RIL	musty, fermented, coffee	1000–10,000 ^c^	0.66 ± 0.05 ^a^	0.49 ± 0.01 ^b^	0.64 ± 0.04 ^a^	0.48 ± 0.02 ^b^
4-methyl-5-thiazoleethanol	2275	MS, RI	nutty	10,800 ^c^	0.20 ± 0.02 ^a^	0.10 ± 0.02 ^b^	0.14 ± 0.05 ^ab^	0.06 ± 0.00 ^c^

Notes: Different letters denote significant differences in the same row (*p* < 0.05). MS, mass spectrum agreed with those of the authentic compound; RI, compound was confirmed by the retention index of the compound standard; RIL, compound was identified according to the retention index from the NIST Chemistry WebBook; N.D., not detected. ^a^ Threshold adapted from An et al. [43]; ^b^ threshold adapted from Czerny et al. [44]; ^c^ threshold adapted from Leffingwell and associates (web pages); ^d^ threshold adapted from Wang et al. [38].

## Data Availability

Data are contained within the article.

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
