# Peer review of "Quality Improvement of Zhayu, a Fermented Fish Product in China: Effects of Inoculated Fermentation with Three Kinds of Lactic Acid Bacteria"

_foods, 2022, doi:10.3390/foods11182756_

Round 1

Reviewer 1 Report

Although the paper by An et al: "Quality improvement of fermented fish products: Effects of inoculated fermentation with three kinds of lactic acid bacteria" is interesting, there are several reasons for doubt regarding the publication. Many mistakes in writing the names of organisms are inadmissible at the academic level. Furthermore, mineral ether? In my opinion, this alone is reason enough to reject the paper, but I ask the authors for a detailed explanation. PCA agar is not a suitable medium for growing LAB because it does not contain enough complex nutrients to get the right results. In my opinion, this is a major experimental failure.

Furthermore, I am asking for full evidence of statistical significance for all results of Figure 2. Only after viewing the results can I judge whether to accept or reject the work. All references should be reviewed and corrected according to the instructions for authors.

Author Response

Dear reviewer,

Thanks so much for your careful review and constructive comments.Those comments are very valuable and helpful for revising and improving our paper. We have studied comments carefully and have made correction which we hope meet with approval. 

The responses were listed in the attached file.

Bests,

Yueqi An

Reviewer 2 Report

In the manuscript "Quality improvement of fermented fish products: Effects of inoculated fermentation with three kinds of lactic acid bacteria," An et al. describe the making of a traditional fermented fish product Zhayu with three LAB starter cultures and analyze the obtained results. The manuscript is well written, but the authors should address several points.

1) Abstract should give a more detailed description of Zhayu, note from which region it originates, and the manuscript's title should also mention this fermented fish product, e.g. "Quality improvement of Zhayu, fermented fish product:..."

2) In the abstract, the authors should provide full Latin names of the species, and TVB-N should not be abbreviated.

3) Why were exactly those three species and strains of LAB used?

4) Please, use the new lab taxonomy (e.g. Lactiplantibacillus plantarum)

5) Define "scoccus"

6) Lactobacillus Plantarum and Streptococcus Pentose > Lactobacillus plantarum and Streptococcus pentose (I'm not aware of the later species; please recheck)

7) Live grass carp (Ctenopharyngodon idellus) (approximately 1500~2000 kg) >> is this the correct weight? Because grass carp should weigh up to 45 kg.

8) Two grams of homogenized and dried samples were extracted with mineral ether rather than diethyl ether. >> is mineral ether misspelling?

9) values of the fermented Zhayu were below 4.5, which were safe for consumption. >> provide a reference or elaborate on the 4.5 limit

10) The addition of LAB increased the acid-producing metabolism, resulting in... the formation of some bacteriostatic substances >> bacteriocin formation was not demonstrated in this work. As such, the authors should remove this claim.

11) Figure 2 legend is incorrect, referencing panels as a, b, c, a, b, c >> please, rectify.

12) Especially Asp and Glu, which showed an umami taste and relatively low thresholds, presented significantly higher contents and TAVs >> when first mentioned in the text, write the full name of the TAV term

13) Tables should be presented in numerical order (1, 2, 3), which they are currently not (1, 3, 2) >> please, rectify

Author Response

Dear Reviewer,

Thanks so much for your careful review and constructive comments.

Those comments are very valuable and helpful for revising and improving our paper. We have studied comments carefully and have made correction which we hope meet with approval.

The responses were listed in the attachment. 

Bests,

Yueqi An

Reviewer 3 Report

This manuscript has evaluated the effects of different LAB starter cultures on the physical, chemical, nutritional and sensory attributes of a Zhayu (fermented fish product). The manuscript is well planned and interesting results are reported.

The manuscript can be accepted after revision;

Comments:

The manuscript lines are not numbered.

Introduction

line 9: change “was” to “is”

Last line: nutritional value.

Materials & methods:

Section 2.3. Sampling line 4: Ginger peels

Section 2.7.2: add a reference for this section

Section 2.7.3: add a reference for this section

2.7.4. Determination of free amino acids line 2: The free amino acids were or The free amino acids extracts were …

Results and discussion:

3.2. Nutritional properties line 2: Figure 2

3.3: texture properties: why the textural properties of different Zhayu samples were different?

This manuscript has evaluated the effects of different LAB starter cultures on the physical, chemical, nutritional and sensory attributes of a Zhayu (fermented fish product). The manuscript is well planned and interesting results are reported.

The manuscript can be accepted after revision;

Comments:

The manuscript lines are not numbered.

Introduction

line 9: change “was” to “is”

Last line: nutritional value.

Materials & methods:

Section 2.3. Sampling line 4: Ginger peels

Section 2.7.2: add a reference for this section

Section 2.7.3: add a reference for this section

2.7.4. Determination of free amino acids line 2: The free amino acids were or The free amino acids extracts were …

Results and discussion:

3.2. Nutritional properties line 2: Figure 2

3.3: texture properties: why the textural properties of different Zhayu samples were different?

Author Response

Dear reviewer,

Thanks so much for your careful review and constructive comments.

Those comments are very valuable and helpful for revising and improving our paper. We have studied comments carefully and have made correction which we hope meet with approval.

The responses were listed in the attached file.

Bests,

Yueqi An

Reviewer 4 Report

The manuscript discusses the use of different strains of lactic acid bacteria and different inoculation rates in improving the quality of fermented fish products. My specific comments are below;

-Please make sections 2.2 and 2.3 more understandable and add a literature that supports the application. Why didn't you use a spectrophotometric method to determine the counts of bacteria? In addition, when you mix the bacteria with fish and other additives at the rate you specify, the counts of bacteria in the product is expected to change. Have you determined this?

-Which type of protein is your target in water soluble proteins? Sarcoplasmic or myofibrils? For which did you determine the ionic strength of the extraction solution?

-Add detailed information for section 2.7.4. The standard used; method (internal or external), stock mix solution concentrations etc.? In addition, detailed information about method validation parameters is required for the reliability of the results.

- Differences depends on strains in results were almost never explained. These parts need improvement.

- Page 7/2: The proteolytic activity of lactic acid bacteria is quite weak. So I think the enzymes promoted here are fish muscle cathepsins. Please rewrite this part by supporting it with literature.

- Page 7/3: Is there a correlation between soluble solid content and pH? Because soluble sugars are used quickly by lactic acid bacteria, the pH may drop. To increase the strength of the manuscript, I suggest performing a correlation analysis between the analysis parameters and explaining the possible relationships.

Author Response

Dear reviewer,

Thanks so much for your careful review and constructive comments.

Those comments are very valuable and helpful for revising and improving our paper. We have studied comments carefully and have made correction which we hope meet with approval.

The responses were listed in the attachment.

Bests,

Yueqi An

Round 2

Reviewer 1 Report

The reviewed manuscript by An et al: "Quality improvement of fermented fish products: Effects of inoculated fermentation with three kinds of lactic acid bacteria" is, after corrections, ready for publication.

Author Response

Point 1: The reviewed manuscript by An et al: "Quality improvement of fermented fish products: Effects of inoculated fermentation with three kinds of lactic acid bacteria" is, after corrections, ready for publication.

Response: Thanks so much for your review and evaluation.

Point 2: English language and style are fine/minor spell check required.

Response: Thank you for your suggestion. In the revision, we checked the spells and improved the English writing.  

Reviewer 4 Report

In general, the authors tried to revise the manuscript by taking into account the referee's suggestions. However, the requested data on method validation parameters (LOD, LOQ, R2, RSD, etc.) used in the analysis of free amino acids in the first report were not presented in the revised manuscript. Presentation of these validation parameters and  chromatogram of standards is important. 

Author Response

Point: In general, the authors tried to revise the manuscript by taking into account the referee's suggestions. However, the requested data on method validation parameters (LOD, LOQ, R2, RSD, etc.) used in the analysis of free amino acids in the first report were not presented in the revised manuscript. Presentation of these validation parameters and  chromatogram of standards is important. 

Response: Thanks so much for your comment. In the revision, we rewrote the methods of the anlysis of free amino acids in detail which we hope meet with approval.
